# Cage Nanofillers’ Influence on Fire Hazard and Toxic Gases Emitted during Thermal Decomposition of Polyurethane Foam

**DOI:** 10.3390/polym16050645

**Published:** 2024-02-27

**Authors:** Arkadiusz Głowacki, Przemysław Rybiński, Monika Żelezik, Ulugbek Zakirovich Mirkhodjaev

**Affiliations:** 1Institute of Chemistry, The Jan Kochanowski University, 25-406 Kielce, Poland; 2Institute of Geography and Environmental Sciences, Jan Kochanowski University, 25-406 Kielce, Poland; monika.zelezik@ujk.edu.pl; 3Department of Biophysics, National University of Uzbekistan, Tashkent 100095, Uzbekistan; u.z.mirkhodjaev@gmail.com

**Keywords:** PUR composites, silsesquioxane, fire hazard, smoke emission, toxicometric index

## Abstract

Polyurethane (PUR), as an engineering polymer, is widely used in many sectors of industries. However, the high fire risks associated with PUR, including the smoke density, a high heat release rate, and the toxicity of combustion products limit its applications in many fields. This paper presents the influence of silsesquioxane fillers, alone and in a synergistic system with halogen-free flame-retardant compounds, on reducing the fire hazard of polyurethane foams. The flammability of PUR composites was determined with the use of a pyrolysis combustion flow calorimeter (PCFC) and a cone calorimeter. The flammability results were supplemented with smoke emission values obtained with the use of a smoke density chamber (SDC) and toxicometric indexes. Toxicometric indexes were determined with the use of an innovative method consisting of a thermo-balance connected to a gas analyzer with the use of a heated transfer line. The obtained test results clearly indicate that the used silsesquioxane compounds, especially in combination with organic phosphorus compounds, reduced the fire risk, as expressed by parameters such as the maximum heat release rate (HRRmax), the total heat release rate (THR), and the maximum smoke density (SDmax). The flame-retardant non-halogen system also reduced the amounts of toxic gases emitted during the decomposition of PUR, especially NOx, HCN, NH_3_, CO and CO_2_. According to the literature review, complex studies on the fire hazard of a system of POSS–phosphorus compounds in the PUR matrix have not been published yet. This article presents the complex results of studies, indicating that the POSS–phosphorous compound system can be treated as an alternative to toxic halogen flame-retardant compounds in order to decrease the fire hazard of PUR foam.

## 1. Introduction

A fire hazard is a result of multiple aspects, including a high released heat value and oxygen deprivation, as well as smoke and toxic gas emissions [1,2].

During a fire hazard, people are exposed not only to fire or smoke but also to a mixture of toxic gases, such as CO, CO_2_, NOx, HCN, SOx, and HCl [3,4,5,6,7].

In the event of a room fire, the most dangerous products of thermal decomposition and combustion are created during the combustion of polyurethane-based foams. Polyurethane foams are used, among others, in the production of upholstered furniture. The research conducted so far indicates that even in the initial phase of the thermal decomposition of PUR foam, the concentrations of CO and CO_2_ are so high that they can pose a lethal threat to people. During the decomposition of PUR foam, high amounts of HCN and NOx, which have been proven to have asphyxiate properties, are also emitted [8,9,10].

Polyurethane, as an engineering composite, is widely used in many sectors of industry. However, the high fire risks associated with PUR, including the smoke density, a high heat release rate, and the toxicity of combustion products, limit its applications in many fields [11,12].

As a consequence, research on how to reduce the flammability and toxicity properties of PUR is attracting more and more attention.

Nowadays, to reduce the fire hazard of PUR composites, an intumescent flame-retardant system, first of all, in the form of expandable graphite (EG), is widely used [13]. EG is halogen free and acts mainly in the condensed phase. Exposed to heat, expandable graphite forms a low-density, thermal insulating layer on the surface of the polymer that prevents the transfer of both heat and oxygen. Unfortunately, the incorporation of EG into the PUR matrix decreases the mechanical properties of PUR, mainly due to its poor adhesion to the PUR matrix and large particle size. Furthermore, the expandable graphite clearly increases smoke emission during the thermal decomposition of PUR composites [14,15,16].

Currently, in order to develop a new intumescent flame-retardant system, organic phosphorous compounds, alone and in synergistic action with silsesquioxanes are being studied. Compounds based on phosphorus have several advantages, such as low toxicity, a lack of release of poisonous halogen gases, and the production of low amounts of smoke during combustion. Phosphorus compounds, similar to graphite, can form a char layer that protects polymers from combustion heat. Unfortunately, the phosphorus compound’s carbon layer created during thermal decomposition very often lacks a homogenous structure. For this reason, the action efficiency of phosphorus compounds in PUR matrices is worse in comparison with expandable graphite. In order to improve the homogenous as well as insulative properties of the carbon layer created during the thermal decomposition of PUR, phosphorus compounds are joined with silsesquioxanes [17].

Silsesquioxanes are polyhedral structures of the general formula (RSiO1.5)n, where R is virtually any organic substituent or a hydrogen atom, and n is an integer, in most cases equal to 6, 8, 10, or 12. The silsesquioxane core is regarded as the smallest obtainable fragment of crystalline silica. The dimension of the silsesquioxane molecule is within 3 nm. This supposes that crystalline silica may stabilize and increase the amount of homogeneous carbon residue after the thermal decomposition of polymer composites [18,19,20,21,22].

Therefore, the aim of this work is to examine the effectiveness of a synergic system, a phosphorus compound with silsesquioxane, in decreasing fire hazards with special consideration for the emission of toxic gases and smoke from PUR composites.

## 2. Materials and Methods

### 2.1. Materials

The subject of this study was polyurethane foam, which was synthesized with the use of polyol (BASF, Elastoflex W5165/140) and isocyanide (BASF, Izo 135/158) (diphenylmethane diisocyanate–MDI) in a 2:1 proportion.

As a flame retardant, an organic phosphorus compound was used, triphenyl phosphate (TPP), which was produced by Everkem, Italy. The phosphorus content was equal to 9.5% (Figure 1).

The silsesquioxanes used were methacrylpropyl POSS MA0735 (POSS MA), amino-propyl POSS AM0265 (POSS AM), and OL POSS OL1170 (POSS OL) (Figure 1).

### 2.2. Preparation of PUR Composites

Flexible polyurethane foams were produced at the laboratory scale using the one-stage method involving a two-component system, with an equivalent ratio of OH to NCO groups equal to 2:1. Component A, i.e., Elastoflex W5165/140, was mixed with the appropriate amount of modifier (5 wt.% POSS and 10 wt.% TPP vs. polyol). Component B was diphenylmethane diisocyanate (MDI) (Table 1).

Components A and B were mixed and poured into a Teflon open mold, and then foam was frothed. Flexible polyurethane foams, after having been left to the end of the growth process for 1 h, were then conditioned to stable mass at a temperature of 23 ± 2 °C and under relative humidity less than 50 ± 5%, according to the PN-EN ISO 291:2010 standard [23].

### 2.3. Methods

#### 2.3.1. Scanning Electron Microscopy

Scanning electron microscopy (SEM) images were assessed by means of a Quanta 250 FEG electron microscope (FEI Company, Hillsboro, OR, USA) with electron gun and field emission (Schottky’s emitter, Hillsboro, OR, USA).

#### 2.3.2. Fourier-Transform Infrared Spectroscopy Analysis

Fourier-Transform Infrared Spectroscopy with attenuated total reflection (FTIR-ATR) was recorded on a PerkinElmer Spectrum (Waltham, MA, USA). Two FTIR spectrophotometer equipped with a single-reflection diamond ATR crystal on a ZeSe plate. Measurements were recorded using a spectrum computer program. The parameters were 4 scans from 400–4000 cm^−1^ in the transmittance mode with a resolution of 4 cm^−1^.

#### 2.3.3. Thermogravimetry Analysis

The thermograms of thermogravimetric analysis (TGA) were recorded using Netzsch STA 449 F3 Jupiter (Selb, Germany) in a temperature range from 25 to 650 °C, for samples 5 ± 1 mg placed in the open Al_2_O_3_ pan. The measurements were performed with a gas flow of 40/20 µL/min in nitrogen/oxygen and a heating rate of 10 °C/min. The results obtained were processed using the Proteus Thermal Analysis 8.0.3 computer program. The parameters considered during thermogravimetric analysis include the temperature at which the sample loses 5% of weight (T_5_), the temperature at which the sample loses 50% of weight (T_50_), the temperature at the maximum rate of sample decomposition (TR_MAX_), the range of the combustion temperature of the residue after thermal decomposition of the sample (∆T_s_) and the residue at a temperature of 600 °C (P_600_)

#### 2.3.4. Toxicity

The studies on the release of the toxic combustion products of PUR composites were carried out using a TG-gas analyzer coupled system (Netzsch TG 209 F1 Libra (Selb, Germany)) coupled with a real-time analyzer Bruker Omega 5 gas analyzer (Billerica, MA, USA). The measurement of toxic gas evolution analysis were performed at a temperature range from 30 °C to 650 °C with a gas flow of 40/20 µL/min in nitrogen/oxygen and a heating rate of 10 °C/min, for the 5 ± 1 mg samples. The gases were analyzed using a FTIR MIR detector (400–4000 cm^−1^) (Billerica, MA, USA). Spectra were registered every 7–8 s (10 scans). Then, the spectra were converted using the Opus GA computer program, version 8.7.41 to the value of the gas emission concentration [ppm]. The gases recorded were CO_2_, CO, HCl, NH_3_, NO and NO_2_.

The emission gas rate in ppm was converted to toxic gases release concentrations in g/m^3^ using the ideal gas law.
PV=nRT
where

*P*—air/gas pressure in atm (1 atm = 1013.25 hPa),

*V*—volume of gas,

*n*—number of particles moles in a gas,

*R*—gas constant (0.08206 $\frac{L\times atm}{mol\times K}$), and

*T*—Kelvin temperature scale (T °C + 273).

By transforming the above formula, the number of moles per liter of air in 1 atm. in 25 °C is converted:$$\frac{n}{V}=\frac{P}{R\times T}=\frac{1\;atm}{298\;K\times 0.08206\;\frac{L\times atm}{mol\times K}}=0.0409\frac{mol}{L}$$

Then, the volume concentration (g/m^3^) of toxic combustion products is converted into mass per volume of air:$$Concentration\frac{g}{m^{3}}=\frac{0.0409\;\times C\left[ppm\right]\times M}{1000}$$
where

*C*—emission gas rate in [ppm], and

*M*—converted gas molar mass.

#### 2.3.5. Microcalorimetry PCFC

The flammability of the PUR, PUR–AM, PUR–MA, PUR–OL, PUR–TPP, PUR–AM–TPP, PUR–MA–TPP and PUR–OL–TPP was tested using PCFC (pyrolysis combustion flow calorimeter) produced by Fire Testing Technology Ltd. (East Grinstead, UK). The procedure was performed in according to the ASTM D 7309 standard [24]. The pyrolizer’s temperature was 650 °C, and the combustor’s heat was 900 °C. The composites were heated at a 1 °C/s rate. The test was performed in conditions of nitrogen/oxygen (80/20 cc/min). During the test, the following parameters were recorded: the heat release rate, the maximum (peak) value of PCFC HRR (W/g), the time to HRR peak, total HR (kJ/g), and the heat release capacity (J/gK).

#### 2.3.6. Flammability

The PUR composites were tested using a cone calorimeter, produced by Fire Testing Technology Ltd., according to the PN-EN ISO 5660 standard [25]. The samples, all having the dimensions 100 × 100 × 50 mm, were tested in a horizontal position with a heat radiant flux density of 35 kW/m^2^. During the test, the following parameters were recorded: initial sample weight, sample weight during testing, final sample weight, time to ignition (TTI), total heat released (THR), effective combustion heat (EHC), the average weight loss rate (MLR), and the heat release rate (HRR).

#### 2.3.7. The Smoke Density

The smoke density was recorded using a Fire Testing Technology Smoke Density Chamber (SDC) according to the PN-EN ISO 5659-2 standard [26]. The samples, all having dimensions 75 × 75 × 15 mm, were tested with a heat radiant flux density of 25 kW/m^2^. During the test, the following parameters were recorded: initial sample weight, weight during the test, final sample weight, maximum specific optical density (D_s_max), specific optical density after 4 min of testing Ds(4), area under the specific optical density curve (VOF4) and light attenuation coefficient after testing.

## 3. Result and Discussion

### 3.1. Surface Morphology of PUR Composites

The test results obtained by SEM clearly indicate that the PUR composites are characterized by a porous structure. Free spaces filled with air during the thermal decomposition of PUR not only increase the efficiency of exothermic combustion reactions through the diffusion of oxygen into the reaction environment, but also catalyze degradation reactions and thermal destruction of PUR chains (Figure 2A,B).

Introducing an organophosphate compound in the form of TPP to the polyurethane foam matrix significantly reduces the size of free spaces in the PUR structure (Figure 2C,D). The results obtained by the SEM analysis clearly indicate that TPP is homogeneously distributed in the polyurethane matrix, and thus limits the degree of porosity of the tested PUR foam.

A significant reduction in the porosity of the tested PUR foam was achieved for the TPP–POSS system. Regardless of the type of POSS used, both the number and size of free spaces in the PUR structure were significantly reduced in relation to the pure PUR as well as in the PUR–TPP composite [27,28].

The analysis of the obtained SEM images clearly indicates that the best effect in terms of reducing the porosity of the PUR foam was achieved for the TPP–OL system. In the case of the PUR−TPP–OL (Figure 2I,J), the porous structure of PUR practically disappeared.

The structural studies of the PUR–POSS composites by the IR spectroscopy revealed the characteristic absorbance bands. The band at 3337 cm^−1^ corresponded to the stretching vibrations of N-H (Figure 3) [29]. The peak at 2868 and 2868 cm^−1^ corresponded to the symmetric and asymmetric stretching vibration of C-H in hydrocarbon chains, respectively. The peak with a maximum of 1710 cm^−1^ derived from stretching vibrations was associated with the occurrence of the carbonyl group. In the IR spectra, the following signals were also visible: signals at 1538 and 1509 cm^−1^ corresponded to the bending vibrations of the N-H group, whereas the signal at 1094 cm^−1^ was associated with the stretching vibrations of C-O-C groups [30,31].

New signals at 1489, 1186 and 959 cm^−1^ were registered for the PUR composites containing TPP. The peak at 1489 cm^−1^ and 959 cm^−1^ corresponded to a vibration aromatic ring, whereas the peak at 1186 cm^−1^ corresponded to the valence vibration of the P-O-C group (Figure 4) [32,33].

Additional signals were not recorded for the FTIR spectrum of the PUR composites containing TPP with POSS (AM, MA, OL). Also, no loss of signals was observed for the spectrum of the PUR−TPP composites. Therefore, it was found that TPP does not react with polyol and isocyanate during the PUR synthesis process, or even POSS compounds added at this stage. 

### 3.2. Thermal Analysis and Flammability of PUR Composites

The results of the thermal analysis indicate that POSS compounds have an ambiguous effect on the thermal parameters of PUR composites (Table 2). The POSS–AM as well as POSS–MA increase the value of the T_5_ and T_50_ parameters, and significantly reduce the value of the dm/dt parameter, especially in the case of POSS–MA (Figure 5A). Limiting the rate of thermal decomposition dm/dt is of key importance from the point of view of reducing the fire hazard of the tested materials. The lower the value of the dm/dt parameter, the lower the intensity decomposition of a given composite, and thus the less flammable composite enters the flame zone.

It is worth noting that the value of the dm/dt parameter was reduced by nearly 18% compared to the reference sample in the case of the PUR–MA composite, which suggests that the intensity of the combustion process (flame feeding) and the emission of toxic gaseous destructs are significantly lower in the case of the POSS–MA sample when compared to the PUR reference sample (Table 2).

All tested POSS compounds (AM, MA, OL) cause an increase in both residues after thermal decomposition, the PTD parameter, as well as residues at a temperature of 600 °C, the P_600_ parameter. A clear increase in the value of the P_TD_ and P_600_ parameters indicates that POSS compounds can catalyze the formation of a ceramic boundary layer during the decomposition of the PUR composite. This layer, due to its insulating properties, primarily limits the transfer of heat from the flame to the interior of the sample, as well as the transfer of liquid and gaseous destructs to the flame [34,35].

The application of an organophosphate flame-retardant compound into the PUR matrix in the form of TPP also had an ambiguous effect on the thermal stability of the PUR–TPP composite (Figure 5B). In the presence of TPP, thermal stability parameters such as T_5_, T_50_ and T_RMAX_ were not improved, while the dm/dt parameter had a value close to the reference sample (PUR composite).

However, it should be noted that the residue parameter after thermal decomposition (P_TD_ parameter) was increased, which indicates that TPP generates a carbon residue of a potential insulating nature during thermal decomposition. The aromatic rings enrich the TPP structure with carbon (Figure 1). Nevertheless, the obtained test results clearly indicate that the carbon residue after thermal decomposition was almost completely burned at ∆T = 435–600 °C. The P600 parameter was only 0.15% in the case of the PUR–TPP composite.

In the case of PUR composites containing the synergistic POSS–TPP system, the amount of residue after thermal decomposition (P_TD_ parameter) was practically no different in relation to the residue after thermal decomposition of the PUR–TPP composite (Figure 5C).

Nevertheless, the P_600_ parameter value for the composites containing the POSS–TPP system was significantly higher when compared to the PUR–TPP composite. These were 4.54, 4.27, and 4.71 for the PUR–AM–TPP, PUR–MA–TPP, and PUR–OL–TPP composites, respectively (Table 2).

The increase in the P_600_ parameter for the composites containing the POSS–TPP system indicates that POSS compounds stabilize the carbon residue through physical interactions, i.e., the formation of a ceramic coating on the carbon surface and as a result of a chemical reaction based on the formation of thermally stable Si-C silicon carbide [36].

It is also worth noting that those composites which contained the POSS–TPP system, especially the PUR–MA–TPP ones, were characterized by higher values of the T_50_ and T_RMAX_ parameters, along with a simultaneously reduced value of the dm/dt parameter, compared to the PUR–TPP reference sample.

The test results obtained using the PCFC method were corelating positively with those obtained when using the thermal analysis method [37,38]. Among the tested POSS compounds, the PUR–MA composite had the highest reduction value for both HRR_MAX_ and THR parameters. It cannot be excluded that despite the lack of changes in the IR spectrum, the POSS–MA was applied to the structure of the PUR foam through polar groups located at the end of the alkyl chains or bound to the PUR through reversible intermolecular interactions [39,40].

POSS–AM and POSS–OL also have a beneficial effect on reducing the flammability of PUR composites; however, the reduction in the values of HRR_MAX_ and THR parameters was observed for a much smaller extent than in the case of the composite containing POSS–MA (Figure 6).

The test results presented in Table 3 indicate that triphenyl polyphosphorus (TPP) is much more effective in reducing the flammability of PUR compared to POSS compounds. In the presence of TPP, the HRR_MAX_ parameter was reduced by 38%. It is also worth noting that the TPP significantly reduced the value of the heat capacity parameter, i.e., HRC. Reducing the HRC parameter value by 37.8% significantly limits the susceptibility of the modified PUR foam to both maintaining combustion processes and fire development (Table 3) (Figure 7).

The highest degree of fire hazard reduction in the tested PUR foams, expressed by means of the HRR_MAX_, THR, and HRC parameter values, was obtained for the synergistic effect of POSS compounds and TPP. The results presented in Table 3 indicate that all tested composites, i.e., PUR–AM–TPP, PUR–MA–TPP and PUR–OL–TPP, are characterized by similar values of HRR_MAX_, THRR_MAX_, THR, and HRC (Figure 8).

Considering that the PUR composites containing the synergistic POSS–TPP system are characterized by the greatest reduction in flammability in comparison to the PUR reference sample, they were next subjected to flammability tests in real conditions (Table 4).

The test results obtained using the cone calorimetry method confirmed the effectiveness of the POSS–TPP system in the flammability reduction processes of the tested PUR foams. In real conditions, it is worth noting that the POSS–TPP system, especially in the form of AM–TPP and OL–TPP, significantly reduced the total heat released (THR parameter) of the tested PUR composites. In the case of AM–TPP, the THR parameter value was reduced by 70.1%, while it was reduced by 68.7% in the case of OL–TPP compared to the reference sample (PUR) (Table 4).

It should also be emphasized that the MA–TPP system effectively reduced the MARHE parameter informing about the intensity of fire. The PUR–MA–TPP composite, similarly to the PUR–OL–TPP one, is characterized by a significant reduction time to flameout compared to the PUR–TPP composite. Reducing the extinguishing time of the sample, without increasing the HRR_MAX_ value, reduces the fire hazard of the tested PUR composites.

### 3.3. Emission of Smoke and Toxic Gases

Smoke is defined as an aerosol of solid or liquid particles generated during the pyrolysis process or thermo-oxidative decomposition of an organic fuel. The amount of smoke generated depends on many factors, including the chemical composition of the fuel, the availability of oxygen, the intensity of the heat flux, as well as the conditions of the combustion process (flameless or flame combustion) [41].

Reducing the amount of smoke emitted directly translates into increased fire safety. It is common knowledge that smoke hinders the orientation of people caught in a fire; and therefore, it is the main factor causing disorientation and panic. It is also a carrier of toxic organic destructs, especially those coming from the group of dioxins and polycyclic aromatic hydrocarbons [42].

The obtained test results clearly indicate that the flame-retardant system used for the tested PUR–POSS–TPP composites has a beneficial effect on reducing their smoke production (Ds_MAX_ parameter, Table 5).

In the case of composites containing AM–TPP and MA–TPP, the value of the maximum optical density of smoke was reduced by over 32% in comparison to the PUR reference (Table 5).

Also, the amount of smoke emitted after 4 min of the test, the Ds(4) parameter, is much lower for the composites containing POSS–TPP than for the PUR reference sample.

It should also be emphasized that the PUR–TPP composite is also characterized by lower values of the Ds_MAX_ and Ds(4) parameters compared to the PUR reference composite. It is probable that the char formed during the thermal decomposition of TPP of good insulating properties limits the decomposition of the composite and reduces smoke emissions.

Silsesquioxanes have a beneficial effect on the reduction in toxic gas emissions generated during PUR decomposition, especially when they occur in a synergistic system with TPP (Table 5) (Figure 9).

Applying the silsesquioxane compounds to the PUR foam matrix results primarily in the reduction in carbon monoxide emissions. It is worth noting that the amount of hydrogen cyanide emitted was also reduced. However, POSS compounds had an ambiguous effect on the CO_2_ and NO_2_ emission values. In the case of the PUR–OL composite, a significant increase in the amount of NO_2_ emitted was recorded.

Applying the organophosphate compound TPP to the PUR matrix results in reduced CO, CO_2_, and HCN emissions compared to the PUR reference sample (Table 6). Nevertheless, TPP, just like POSS–OL, clearly affects the amount of nitrogen dioxide (NO_2_) emissions. Considering that both TPP and POSS–OL do not contain nitrogen, it is possible that TPP and POSS–OL accelerate the oxidative decomposition of the urethane bond.

The synergistic POSS–TPP system clearly reduces the amount and toxicity of gases emitted during the decomposition of polyurethane composites. The POSS–TPP system not only reduces the amount of CO and CO_2_, but also the amount of HCN and NO_2_ emissions during the decomposition of PUR composites (Figure 10).

The reduction in toxic gas emissions in the POSS–TPP system results primarily from the formation of a uniform carbon layer during the thermal decomposition of the PUR composite. The main sources of carbon in the PUR–POSS–TPP composite are the aromatic structures of TPP, which, during thermal decomposition of the composite, create a three-dimensional carbon residue with a large specific surface area, also chemically stabilized by the POSS decomposition products. The reduction in the emission values of the determined toxic gases results primarily from their impeded diffusion through the boundary layer, as well as the effect of adsorption processes taking place in the boundary layer.

## 4. Summary

This paper presented the influence of silsesquioxane fillers, along with a synergistic system and halogen-free flame-retardant compounds, on reducing the fire hazard of polyurethane foams.

The test results obtained by the SEM clearly indicate that the PUR composite is characterized by a porous structure. The best effect in terms of reducing the porosity of the PUR foam was achieved for the TPP–OL system. In the case of the PUR−TPP–OL system, the porous structure of PUR practically disappeared.

The thermal analysis results indicated that POSS compounds have an ambiguous effect on the thermal parameters of PUR composites.

All the tested POSS compounds (AM, MA, and OL) cause an increase in both residues after the thermal decomposition of the PTD parameter, as well as residues at a temperature of 600 °C when it comes to the P_600_ parameter. A clear increase in the value of the P_TD_ and P_600_ parameters indicates that POSS compounds can catalyze the formation of a ceramic boundary layer during the decomposition of the PUR composite.

The P_600_ parameter value for the composites containing the POSS–TPP system is significantly higher when compared to the PUR–TPP composite. The increase in the P_600_ parameter for the composites containing the POSS–TPP system indicates that POSS compounds stabilize the carbon residue through physical interactions, i.e., the formation of a ceramic coating on the carbon surface and because of a chemical reaction based on the formation of thermally stable Si-C silicon carbide.

The PUR–MA composite has the highest reduction value for both HRR_MAX_ and THR. It cannot be excluded that despite the lack of changes in the IR spectrum, the POSS–MA was applied to the structure of the PUR foam through polar groups located at the end of the alkyl chains or bound to the PUR through reversible intermolecular interactions.

The obtained test results clearly indicate that the flame-retardant system used for the tested PUR–POSS–TPP composites has a beneficial effect on reducing their smoke production.

Silsesquioxanes have a beneficial effect on the reduction in toxic gas emissions generated during PUR decomposition, especially when they occur in a synergistic system with TPP. 

The synergistic POSS–TPP system clearly reduces the amount and toxicity of gases emitted during the decomposition of polyurethane composites. The POSS–TPP system reduces not only the amount of CO and CO_2_, but also the amount of HCN and NO_2_ emissions during the decomposition of PUR composites.

## Figures and Tables

**Figure 1 polymers-16-00645-f001:**
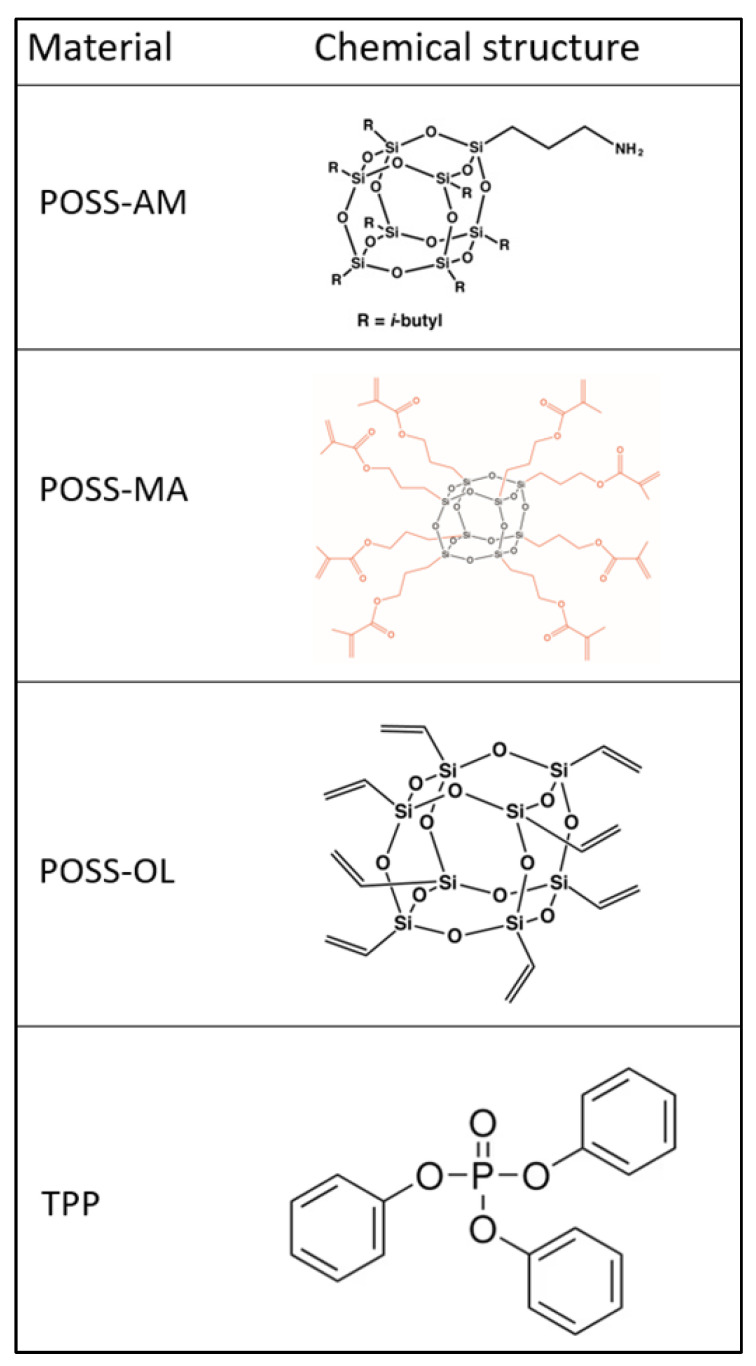
Chemical structure of used compounds.

**Figure 2 polymers-16-00645-f002:**
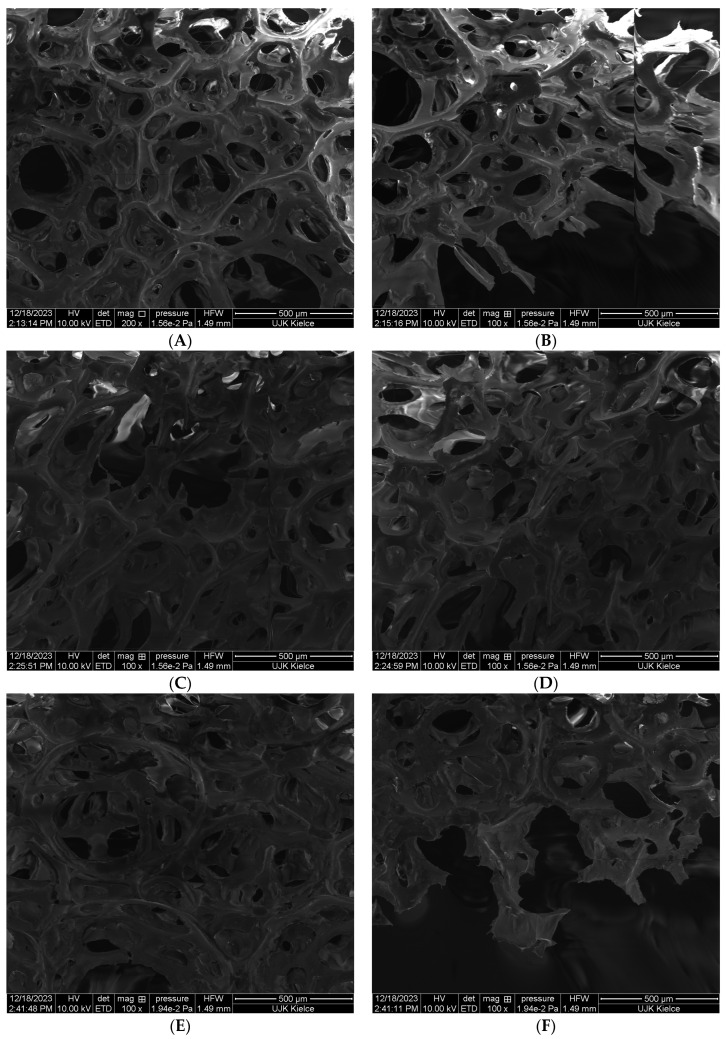
SEM images of PUR composites: (**A**,**B**) reference PUR composite; (**C**,**D**) PUR composite containing TPP; (**E**,**F**) PUR composite containing an AM–TPP system; (**G**,**H**) PUR composite containing an TPP–MA system; (**I**,**J**) PUR composite containing an AM–TPP OL–TPP system.

**Figure 3 polymers-16-00645-f003:**
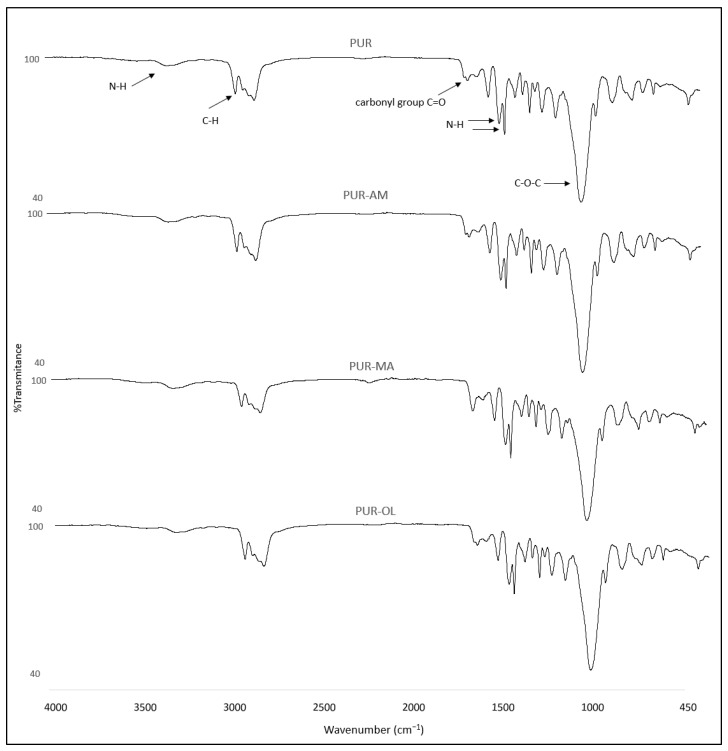
FTIR spectra of PUR, PUR–AM, PUR–MA and PUR–OL composites.

**Figure 4 polymers-16-00645-f004:**
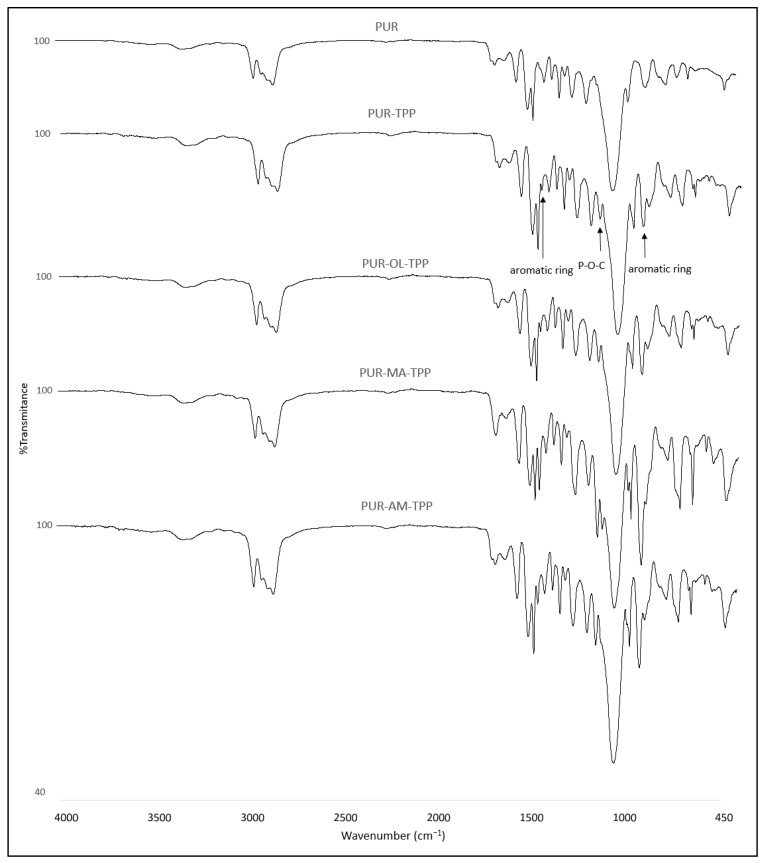
FTIR spectra of the PUR, PUR–TPP and PUR–OL–TPP composites.

**Figure 5 polymers-16-00645-f005:**
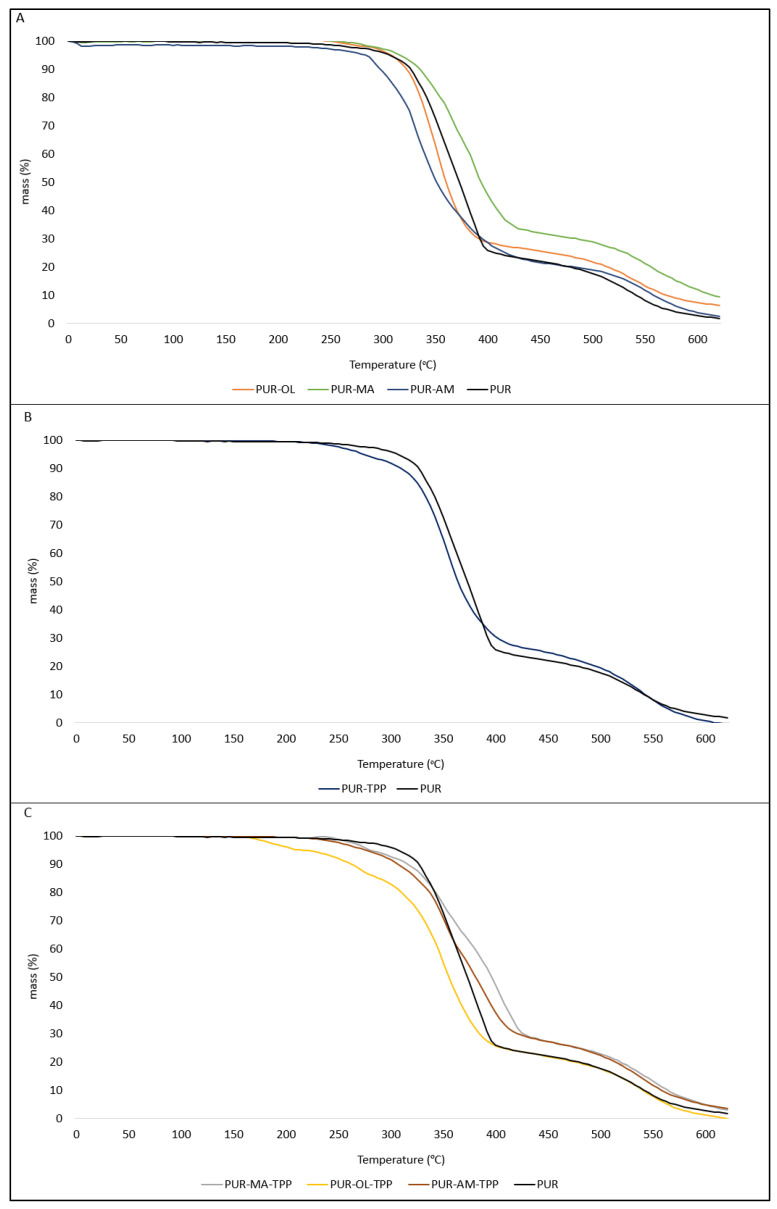
TG thermograms of PUR composites: (**A**) PUR, PUR–AM, PUR–MA, and PUR–OL; (**B**) PUR and PUR–TPP; (**C**) PUR, PUR–AM–TPP, PUR–MA–TPP, and PUR–OL–TPP.

**Figure 6 polymers-16-00645-f006:**
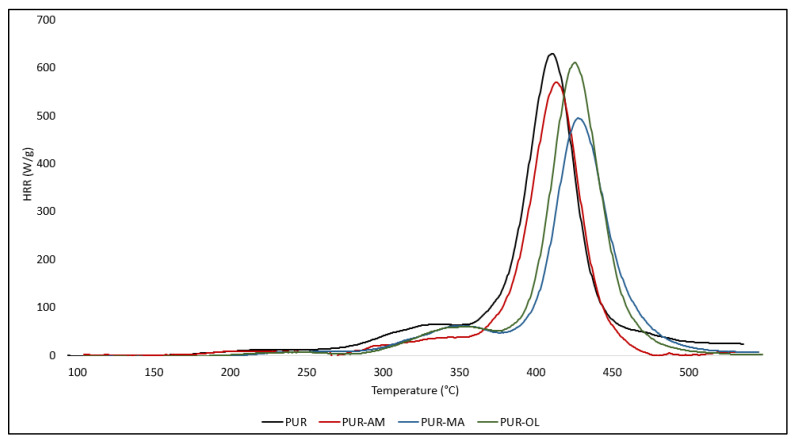
HRR curves of PUR composites: PUR, PUR–AM, PUR–MA, and PUR–OL.

**Figure 7 polymers-16-00645-f007:**
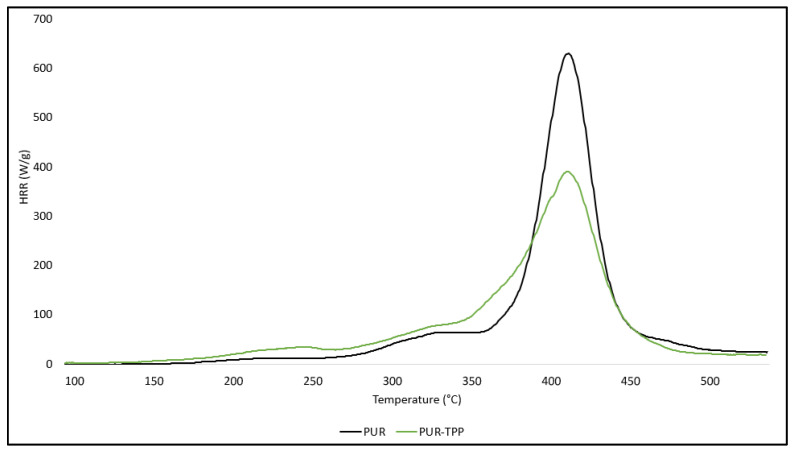
HRR curves of PUR composites: PUR, PUR–TPP.

**Figure 8 polymers-16-00645-f008:**
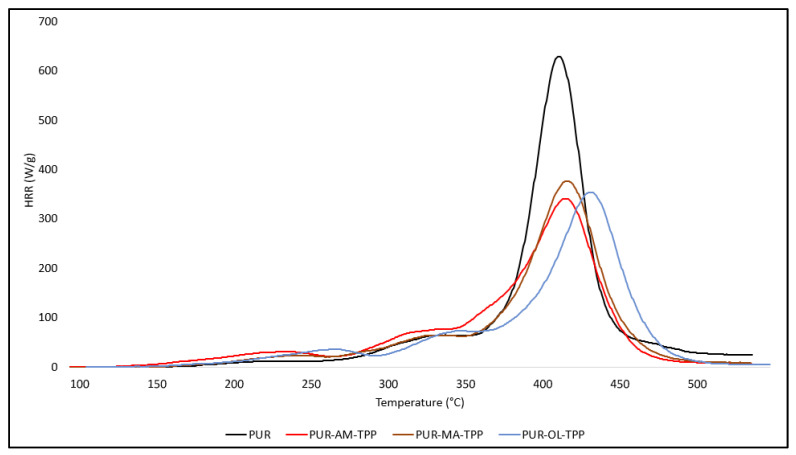
HRR curves of PUR composites: PUR, PUR–AM–TPP, PUR–MA–TPP, and PUR–OL–TPP.

**Figure 9 polymers-16-00645-f009:**
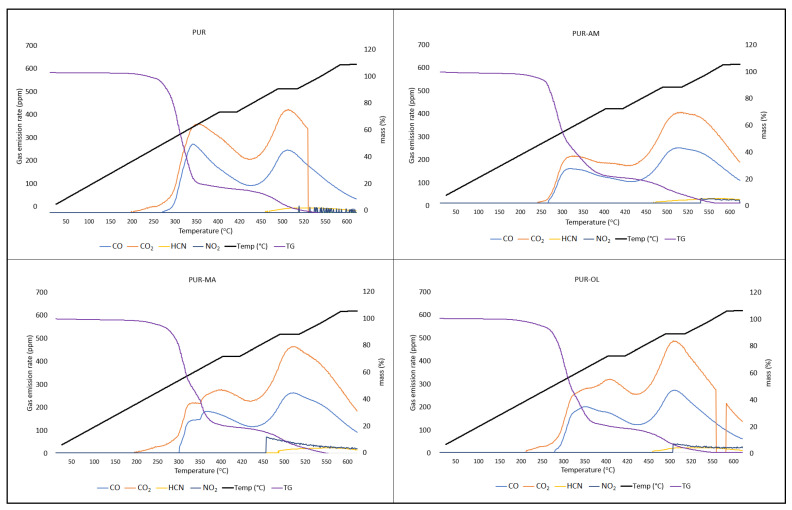
TG-gas analyzer—real-time gas emission results for PUR, PUR−AM, PUR−MA, and PUR−OL.

**Figure 10 polymers-16-00645-f010:**
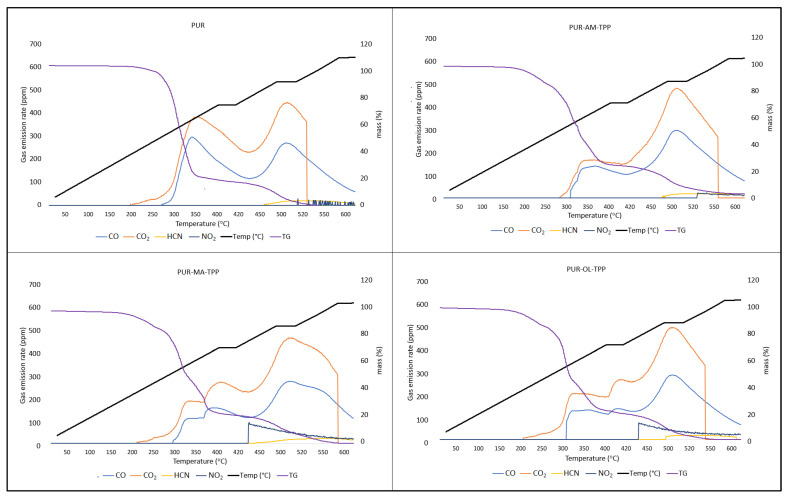
TG-gas analyzer—real-time gas emission results of PUR, PUR−AM−TPP, PUR−MA−TPP, and PUR−OL−TPP.

**Table 1 polymers-16-00645-t001:** Composition of PUR composites in mass fraction (phr).

Composite	AM	MA	OL	TPP
PUR–AM	5			
PUR–MA		5		
PUR–OL			5	
PUR–TPP				10
PUR–AM–TPP	5			10
PUR–MA–TPP		5		10
PUR–OL–TPP			5	10

**Table 2 polymers-16-00645-t002:** Thermal analysis results of PUR composites.

Composite	T_5_ (°C)	T_50_ (°C)	T_RMAX_ (°C)	dm/dt (%/min)	P_TD_ (%)	∆T_s_ (°C)	P_600_ (%)
PUR	295	365	365	10.1	20.1	460–600	1.45
PUR–AM	271	345	330	9.3	23.7	475–600	3.60
PUR–MA	305	385	370	8.3	31.6	475–600	10.81
PUR–OL	295	350	340	10.6	27.2	450–600	7.20
PUR–TPP	260	350	345	9.6	25.5	235–600	0.15
PUR–AM–TPP	270	375	360	6.7	26.6	450–600	4.54
PUR–MA–TPP	270	385	395	6.1	25.2	465–600	4.27
PUR–OL–TPP	195	345	345	7.4	23.6	455–600	4.71

**Table 3 polymers-16-00645-t003:** Flammability test results of PUR composites obtained with the use of the PCFC method.

Composite	HRR_MAX_ (W/g)	THRR_MAX_ (°C)	THR (kJ/g)	HRC (J/gK)
PUR	629.0	415	26.3	578
PUR–AM	570.5	410	22.8	532
PUR–MA	495.1	420	23.1	461
PUR–OL	610.1	415	26.7	578
PUR–TPP	390.1	415	24.8	354
PUR–AM–TPP	341.1	420	24.3	315
PUR–MA–TPP	375.5	420	23.5	349
PUR–OL–TPP	354.4	420	24.0	331

**Table 4 polymers-16-00645-t004:** Flammability test results of PUR composites obtained using the cone calorimeter method.

Composite	PUR	PUR–TPP	PUR–AM–TPP	PUR–MA–TPP	PUR–OL–TPP
t_i_ (s)	30	40	41	79	32
t_f-o_ (s)	944	1098	1094	205	315
HRR (kW/m^2^)	62.70	43.46	109.9	47.77	67.4
HRR_max_ (kW/m^2^)	145.92	97.42	162.1	101.3	131.7
tHRR_max_ (s)	115	85	120	145	100
THR (MJ/m^2^)	60.5	46.1	17.9	49.0	18.9
EHC (MJ/kg)	17.25	13.61	14.10	13.81	13.8
EHC_max_ (MJ/kg)	67.35	72.60	74.31	74.55	65.28
MLR (g/s)	0.032	0.028	0.069	0.031	0.043
MLR_max_ (g/s)	0.151	0.122	0.204	0.102	0.132
AMLR (g/m^2^·s)	3.74	3.51	12.27	4.34	9.15
FIGRA (kW/m^2^·s)	1.27	1.14	1.35	0.69	1.317
MARHE (kW/m^2^)	81.55	56.89	93.13	51.46	80.15

**Table 5 polymers-16-00645-t005:** Smoke emission of PUR composites.

Composite	D_sMAX_	TD_sMAX_	Ds(4)	VOF_4_
PUR	300.1	600	109.4	211.3
PUR–TPP	263.5	600	107.2	217.3
PUR–AM–TPP	203.8	600	74.22	153.3
PUR–MA–TPP	203.4	600	96.29	188.7
PUR–OL–TPP	212.9	600	89.12	202.9

**Table 6 polymers-16-00645-t006:** Concentration of toxic gases emitted during thermal oxidation decomposition of PUR composites.

Composite	Gas Concentration (g/m^3^)				
	CO	CO_2_	HCN	NO_2_	Total
PUR	71.52	182.28	2.22	0.972	256.9
PUR–AM	68.49	175.67	1.89	1.054	247.0
PUR–MA	69.20	199.50	2.04	1.412	272.1
PUR–OL	69.21	192.64	2.15	6.03	269.9
PUR–TPP	66.65	154.86	2.04	14.03	237.5
PUR–AM–TPP	64.90	145.40	1.98	11.24	238.4
PUR–MA–TPP	69.51	154.67	1.94	10.37	236.5
PUR–OL–TPP	64.19	161.18	1.74	14.47	241.5

## Data Availability

Data are contained within the article.

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
