# Peer review of "Cage Nanofillers’ Influence on Fire Hazard and Toxic Gases Emitted during Thermal Decomposition of Polyurethane Foam"

_polymers, 2024, doi:10.3390/polym16050645_

Round 1
Reviewer 1 Report
Comments and Suggestions for Authors
I would like to thank the authors for a useful manuscript, which deals with a very interesting and current topic. The article is focused on the reduction of flammability of polyurethane foams and overall issues related to their combustion. The manuscript provides interesting information on the use of the synergistic system of POSS and TPP and tests a complex matrix of materials. Some of the conclusions based on the series of analyses are obvious, but some are less clearly described from my point of view and could be better defined, commented and possibly added. The weaknesses are discussed in more detail below.
You mention "PCFC microcalorimeter (pyrolysis combustion flow calorimeter)" in the abstract - I would rather say: pyrolysis combustion flow calorimeter (PCFC) or micro calorimetry by pyrolysis combustion flow calorimeter (PCFC).
You list abbreviations in the abstract that have not been explained (HRRmax, THR or SDmax) - either give the full terms or the full terms together with the abbreviations. I would not replace the continuous explanation of abbreviations in the text with a list of abbreviations at the end of the manuscript.
Line 12 states "Polyurethane as one of engineering composite" - polyurethane is not a composite but a polymer. I would unify the terminology and meaning of polyurethane-based foams from the beginning.
On line 13, the abbreviation PUR is given, which is not previously assigned to polyurethane. The abbreviation should always be explained as polyurethane (PUR) first and then can be used separately.
On lines 32 and 33 you list a number of chemical compounds in full names and it would be appropriate, in my opinion, to mention them here with chemical formulas as in the abstract. I would have used a subscript for NH3 and CO2 in the abstract. Please check the subscripts for chemical compounds and units throughout the manuscript (e.g., also line 137).
Merge the citations (e.g., on line 33 and 40) as follows [3-7] or [8-10] - there are several such occurrences in the manuscript.
The quality of Figure 1 is very low and I would ask you to redraw it or insert it in a higher quality.
On lines 91 and 92, you state “Component B was MDI, diphenylmethane diisocyanate” - I would say Component B was diphenylmethane diisocyanate (MDI).
On lines 178 and 179 you state "Introducing an organophosphate compound in the form of TPP into the polyurethane foam matrix, significantly reduces the size of free spaces in the PUR structure" - On what basis do you state that there is a significant reduction in the free space volume of the PUR foam? From my point of view, there is not a significant difference between the images (Figure 2 C,D,...). Do you have the options to determine the specific surface area and pore size using Brunauer-Emmett-Teller (BET) analysis? If not, I would at least focus on finding the areas where the differences are really obvious and brighten the images. Furthermore, it is not entirely clear to me why you list a total of 10 images when a total of eight materials were tested. Please also unify the labelling of the materials (in Table 1 you list OL-TPP and in the caption of Figure 2 you list TPP-OL).
The parameters T5 and T50 should already be defined in section 2.3.3 and I would also include the parameters PTD, P600 as mass residue at 600 °C, etc. Please keep the subscripts. From my point of view, it would be useful to graphically indicate all parameters in a TGA plot.
It should be clear from Table 1 what the numbers 5 and 10 (mass fractions) mean.
The title of chapter 2.3.1 should be Scanning electron microscopy and you would define the abbreviation SEM later in the text. Similarly, I would not use the abbreviation FTIR in the title of the next chapter. I would not put the abbreviation TGA in brackets in the title of Chapter 2.3.3, but would put it later in the text. It is not necessary, but I would have a consistent style for the headings.
For TGA you mention sample sizes "for samples 4-6 mg" and for toxicity you mention "for 5 mg ± 0.5 mg samples" - please standardise the notation of sample sizes (either interval or deviation).
Rather than the formulation “PUR composited” on line 153, please list PUR composites.
Chapter "1. Surface morphology of PUR composites" should be labelled 3.1. ... Please adjust the numbering of the other chapters to be consistent throughout the manuscript. The numbering of the chapter Thermal analysis and flammability of PUR composites (line 211) is also incorrect.
You give numerical values with a decimal point in the text when evaluating results from FTIR analysis - always use a decimal point and I would use integer values for wavenumbers in this case.
Do not mention Fig. 3 in the text when you use Figure 3 in the figure caption - it would be better to unify.
I would not include a bold box for all Figures. For Figure 3, I would give the caption: FTIR spectra of PUR and composites PUR-AM, PUR-MA and PUR-OL. Similarly, I would distinguish PUR and composites throughout the manuscript. Units should always be in round brackets and not in square brackets.
It would be correct to include spectra for PUR-TPP-AM and -MA in Figure 4. The spectra could be closer together so that Figures 3 and 4 are not so large.
I am not an expert in English, but in the text "The lower the value of the dm/dt parameter, the lower the intensity" already seems to me too many definite articles.
Tables 1 and 2 could be narrower and all tables could have the values centered in the columns.
The caption of Figure 5 should not be on a new page. What does the TG (%) parameter mean? I would expect more like mass (%). You use once TGA and once TG for Thermogravimetry analysis - unify.
Add % for the P600 parameter values on lines 253 and 254.
In Figure 6 the curves are quite hard to distinguish - please choose different curve colours. It would be advisable to always choose the same curve colour for a specific material.
Why are the results from the cone calorimeter for PUR-AM, PUR-MA and PUR-OL not shown? Many of the parameters reported in Table 4 are not explained in the manuscript and may not be clear to all readers who do not use this analytical method.
The description of the axes in Figures 9 and 10 is very difficult to read - please enlarge the font. Also, there is no need to show the y-axis values with two decimal places and it is not clear when the main and second y-axis lines are not shown.
It would be advisable to add at least one representative photo of the sample after the combustion test to the manuscript to show the homogeneous carbon layer, ceramic coating on the carbon surface.
Author Response
Responses on the remarks reviewer no. 1 are in the attachment

Reviewer 2 Report
Comments and Suggestions for Authors
The flame retardant composites were prepared with different structures of AM, MA and OL monomers and polyurethane resins. The properties of the composites were characterized by thermogravimetric analysis, SEM, and cone calorimetry. This manuscript cannot be published in this journal until it has been major revision. The suggestions are as follows:
1. The abbreviation that first appears needs to be defined, such as PUR in the abstract. In addition, there is a lack of characterization values in the abstract, and some gas indices need to be improved, such as NH3 and CO2 in the abstract.
2. Some paragraphs in the manuscript need to be merged or expanded. In addition, the introduction section needs to be improved, and at the same time, multiple aspects of research on flame retardancy should be provided as evidence in the introduction.
3. How to determine whether TPP is uniformly distributed in the polyurethane matrix based on SEM?
4. Why choose POSS-AM, POSS-MA, and POSS-OL for the experiment? This should be explained in the manuscript.
5. The infrared spectra of PUR are repeatedly listed, and additional infrared spectra of PUR-TPP-AM and PUR-TPP-MA need to be added.
6. Please explain why PUR-MA has such a high thermal decomposition temperature at T5 (305℃), and why PUR-OL-TPP has such a low thermal decomposition temperature at T5 (195℃)?
7. The manuscript should focus on analyzing the differences in flame retardant performance after adding AM, MA, and OL.
Comments on the Quality of English LanguageEnglish needs improvement.
Author Response
Responses on the remarks reviewer no. 2 are in the attachment

Round 2
Reviewer 1 Report
Comments and Suggestions for Authors
I thank to the authors for revision of the manuscript according to my comments. Most of the weaknesses which I pointed out have been corrected and I currently consider the manuscript acceptable for publication in this journal. For future research on these materials, I would focus on a more detailed characterization using SEM with at least basic statistical analysis, and I would try other methods to evaluate the porosity of the material, even if it is a limited volume analysis and only open pores are evaluated. When you directly state the evaluation of surface morphology, I still think the BET method would be appropriate. I would also recommend implementing of the photographic documentation and, as the above comment suggests, supporting the condition of the sample after a certain stage of testing.
Reviewer 2 Report
Comments and Suggestions for Authors
The current version can be accepted.
Comments on the Quality of English LanguageThe English expression needs to be slightly revised.